# A Semi-Automatic Approach for Holistic 3D Assessment of Temporomandibular Joint Changes

**DOI:** 10.3390/jpm13020343

**Published:** 2023-02-16

**Authors:** Michael Boelstoft Holte, Henrik Sæderup, Else Marie Pinholt

**Affiliations:** 13D Lab Denmark, Department of Oral and Maxillofacial Surgery, University Hospital of Southern Denmark, Finsensgade 35, 6700 Esbjerg, Denmark; 2Department of Regional Health Research, Faculty of Health Sciences, University of Southern Denmark, Finsensgade 35, 6700 Esbjerg, Denmark

**Keywords:** three-dimensional imaging, cone-beam computed tomography, computer-assisted surgery, temporomandibular joint, orthognathic surgery, holistic assessment

## Abstract

The literature lacks a reliable holistic approach for the three-dimensional (3D) assessment of the temporomandibular joint (TMJ) including all three adaptive processes, which are believed to contribute to the position of the mandible: (1) adaptive condylar changes, (2) glenoid fossa changes, and (3) condylar positional changes within the fossa. Hence, the purpose of the present study was to propose and assess the reliability of a semi-automatic approach for a 3D assessment of the TMJ from cone-beam computed tomography (CBCT) following orthognathic surgery. The TMJs were 3D reconstructed from a pair of superimposed pre- and postoperative (two years) CBCT scans, and spatially divided into sub-regions. The changes in the TMJ were calculated and quantified by morphovolumetrical measurements. To evaluate the reliability, intra-class correlation coefficients (ICC) were calculated at a 95% confidence interval on the measurements of two observers. The approach was deemed reliable if the ICC was good (>0.60). Pre- and postoperative CBCT scans of ten subjects (nine female; one male; mean age 25.6 years) with class II malocclusion and maxillomandibular retrognathia, who underwent bimaxillary surgery, were assessed. The inter-observer reliability of the measurements on the sample of the twenty TMJs was good to excellent, ICC range (0.71–1.00). The range of the mean absolute difference of the repeated inter-observer condylar volumetric and distance measurements, glenoid fossa surface distance measurements, and change in minimum joint space distance measurements were (1.68% (1.58)–5.01% (3.85)), (0.09 mm (0.12)–0.25 mm (0.46)), (0.05 mm (0.05)–0.08 mm (0.06)) and (0.12 mm (0.09)–0.19 mm (0.18)), respectively. The proposed semi-automatic approach demonstrated good to excellent reliability for the holistic 3D assessment of the TMJ including all three adaptive processes.

## 1. Introduction

Three-dimensional (3D) assessment of postoperative condylar resorption and remodeling following orthognathic surgery using superimposed cone beam computed tomography (CBCT) imaging has been studied and published numerous times [1,2,3]. In contrast, only a single recent study has been conducted assessing 3D glenoid fossa changes following orthognathic surgery [4]. The study found significant glenoid fossa changes, and subjects with postoperative condylar resorption showed a significantly higher degree of morphological change in the anterior glenoid fossa than subjects without [4].

Studies suggest 3D superimposition methods to assess glenoid fossa changes in growing subjects [5,6], and to assess the joint space [7,8]. However, none of the applied methods have been validated. Superimposition of CBCT enables automation of 3D dentomaxillofacial assessment and visualization without the need for cephalometric landmark re-identification [1,9,10,11]. Surface-based registration (SBR) comprises the approximation of two surfaces by selection and alignment using the iterative closest point algorithm [12]. Voxel-based registration (VBR) [13,14] uses the grayscale of the voxels to align two 3D volumetric images to the best superimposition [15]. It is today’s method of choice [12,16,17,18] and is considered more reliable than landmark-based cephalometric analysis [16]. Alignment by VBR, validated with the anterior cranial base as a reference, shows high accuracy [19], and is considered to be more consistent than SBR [12,19]. However, a recent study by Holte et al. [20], validated and compared VBR and SBR on the mandibular rami for long-term 3D evaluation of condylar remodeling following orthognathic surgery. In this particular case, SBR was found to be more accurate and more reliable than VBR, possibly due to the influence of mandibular ramal remodeling [20].

Three adaptive processes in the temporomandibular joint (TMJ) are believed to contribute to the position of the mandible: (1) adaptive condylar changes, (2) glenoid fossa changes, and (3) condylar positional changes within the fossa [21]. Adaptive morphovolumetrical and positional changes of the TMJ are thought to be induced during mastication or mechanical loading, e.g., by orthodontic treatment, orthognathic surgery, trauma, or general disease such as rheumatoid arthritis, where the condyles may be displaced in the glenoid fossa [22].

In order to study and understand the postoperative stability and functionality following orthognathic surgery, the focus should not solely be on the condylar changes, but importantly, on the change of the entire TMJ, involving all three adaptive processes. No studies have been identified, which propose a validated holistic approach for 3D assessment of the TMJ including all three adaptive processes. However, the accuracy of the 3D assessment of condylar changes from CBCT has been studied previously and the assessment was shown to be highly accurate for both volumetric and surface discrepancy measurements [23]. Similarly, superimposition on the cranial base and the mandibular ramus has shown high accuracy [19,20]. Hence, the purpose of the present study was to propose and assess the reliability of a semi-automatic approach for holistic 3D assessment of the TMJ from CBCT following orthognathic surgery, including condylar and glenoid fossa changes, as well as the interrelated positional changes. The null hypothesis was: H0: it is not possible to develop a reliable holistic approach for 3D assessment of the TMJ. The approach was deemed reliable if the intra-class correlation coefficient (ICC) was good (>0.60), according to Cicchetti [24].

## 2. Materials and Methods

### 2.1. Study Sample

This study was based on pre- and postsurgical (two years) CBCT-scans from a study sample diagnosed with maxillary and/or mandibular growth disturbances, who underwent a combined bilateral sagittal split osteotomy (BSSO) and Le Fort I procedure at the Department of Oral and Maxillofacial Surgery, University Hospital of Southern Denmark, Esbjerg, Denmark. The study sample was selected, such that half of the subjects were diagnosed with postoperative condylar resorption and the other half without.

Inclusion criteria: age range of 18–65 years; diagnosis indicating a combined BSSO and Le Fort I osteotomy; availability of patients’ pre- and postoperative (two years) CBCT scans. Exclusion criteria: previous history of oral and maxillofacial surgery, presence of craniofacial anomaly syndrome, rheumatoid arthritis, or previous trauma.

### 2.2. Image Acquisition

The CBCT-images were acquired using an i-CAT scanner, version 17–19 (Imaging Sciences International, Hatfield, PA: 120 kVp; 5 mA; 7 s exposure time; Field-of-View (FOV) = 23 × 17.8 cm (768 × 768 × 576 voxels)); isotropic voxels of 0.30 mm. The subjects were scanned in an upright natural head position. The condyles were seated in centric relation and the preoperative occlusion was fixated by a wax-bite, however, without a wax-bite at the postoperative CBCT scanning acquisition. The CBCT data were exported in DICOM format and imported into Mimics^®^ 24 (Materialise NV, Leuven, Belgium).

### 2.3. Surgical Technique

The surgery was performed with general anesthesia as a mandible-first procedure. After BSSO, the distal mandibular segment was positioned using an intermediate splint and was fixated bilaterally to each proximal segment with two 2.0 four hole osteosynthesis plates (KLS Martin, Tuttlingen, Germany). Following the Le Fort I osteotomy, the maxilla was segmented into three pieces, in segmental cases, and the tooth-bearing segments were positioned into the final occlusion without the use of a splint, and subsequently fixated using two 2.0 Y-plates anteriorly and two L-plates posteriorly (KLS Martin, Tuttlingen, Germany). All osteotomy sites were grafted.

### 2.4. Holistic Assessment of the TMJ

The holistic assessment of the TMJ including all three adaptive processes (Figure 1) was performed using Mimics^®^ 24.0 and 3-matic^®^ 16.0 (Materialise NV, Leuven, Belgium). The workflow was automated using Python scripting, Python 3.8 (Python Software Foundation, Fredericksburg, Virginia), and semi-automation was achieved using superimposition techniques as shown in Figure 2 and described in the following.

#### 2.4.1. Assessment of Condylar Changes

The pre- and postoperative mandibles were semi-automatically segmented according to Holte et al. [20]. Subsequently, the segmentation of the mandibular condyle was refined and registration was automatically performed to align the right and left pre- and postoperative rami, separately, as proposed in the protocol validated by Verhelst et al. [23]. Surface-based registration was used for the alignment, which was shown to be more accurate and reliable than VBR for the assessment of mandibular condyle remodeling [20].

The C-plane, centered in the pre-operative mandibular notch (C-point) and parallel to the Frankfurt horizontal plane, as defined by Xi et al. [25], was used to isolate the pre- and postoperative condyles from the rami. For spatial analysis, the condylar head was spatially divided into four sub-regions: anterior/posterior-lateral and medial condylar head using two cutting planes: (1) a plane going through the lateral and medial condylar poles perpendicular to the horizontal Frankfurt plane, and (2) an orthogonal mid-plane defined by the lateral and medial condylar poles. Postoperative change of the condyle was represented by the volumetric change and by color-coded distance maps [14], quantified by the mean surface distance of the condylar head and neck, and the defined four sub-regions of the condylar head (Figure 1).

#### 2.4.2. Assessment of Glenoid Fossa Changes

The pre- and postoperative crania were semi-automatically segmented according to Holte et al. [4] For alignment of the segmented pre- and postoperative crania, the postoperative CBCT scan was registered to the preoperative CBCT scan by VBR using the anterior cranial base, zygomatic arches, and forehead as the volume of interest unaffected by the surgery [19]. Next, a curve was manually traced engulfing the preoperative glenoid fossa, which was automatically attracted and attached to the surface of the postoperative skull, defining the postoperative glenoid fossa (Figure 1). A three-dimensional assessment of glenoid fossa changes was performed according to Holte et al. [4]. However, instead of applying the midsagittal and coronal plane for spatial division of the glenoid fossa through its center of gravity [4], the glenoid fossa was divided analogously to the division of the condylar head into four sub-regions for spatial analysis using the two previously defined cutting-planes. Glenoid fossa surface discrepancies were represented by color-coded distance maps, [14] and quantified by the root mean square (RMS) surface distance of the total glenoid fossa and the four defined fossa sub-regions (Figure 1) [4].

#### 2.4.3. Assessment of Interrelated Positional Change in the Joint Space

The 3D assessment of interrelated positional change in the joint space was calculated as the pre- to postoperative change in the minimum distance between the mandibular condyle and the glenoid fossa for each of the four sub-regions (Figure 1).

### 2.5. Statistical Analysis

Statistical analysis of the data was performed in STATA® 16.1 (StataCorp, College Station, TX, USA). A sample size calculation for a one-sample correlation study (ICC > 0.6, power = 0.8, alpha = 0.05, and two raters) was performed. To evaluate the reliability, two observers independently performed the assessment (MBH, HS). The inter-observer reliability of the measurements was summarized using the mean absolute difference (MAD) and standard deviation (SD), similar to related reliability studies [9,10,11,12,20,23]. Intraclass correlation coefficients at a 95% confidence interval on measurements of the two observers were calculated for single measurements using a one-way random effect model. Bland-Altman plots were produced to evaluate observer agreement with 95% limits of agreement.

## 3. Results

The statistical sample size calculation resulted in a required sample size of n = 20 fossae to obtain a statistical power of 0.8. Hence, ten post-pubertal patients in a database (March 2012–November 2017) who met the inclusion- and exclusion criteria, were included. Nine female; one male; mean age 25.6 ± 6.9 years; skeletal class II malocclusion with mandibular retrognathia (eight subjects); maxillary retrognathia (two subjects); anterior open bite (nine subjects); mandibular asymmetry (three subjects); treated with maxillomandibular advancement (ten subjects); maxillary expansion (ten subjects) and with genioplasty (two subjects).

The inter-observer ICCs MADs and SDs are presented in Table 1 and Table 2. The inter-observer ICCs for the volumetric and mean surface condylar measurements, glenoid fossa RMS distance measurements, and change in minimum joint space distance measurements were all excellent: 1.00, 0.99, 0.82 and 0.91, respectively. The inter-observer MAD (SD) were 1.68% (1.58), 0.10 mm (0.14), 0.06 mm (0.04) and 0.17 mm (0.16), respectively. For 3D assessment of the condylar head and neck the ICCs were excellent, (0.99–1.00) and (0.96–0.98), with inter-observer volumetric MAD (SD) values of 2.71% (2.73) and 2.26% (2.04), respectively, and with inter-observer mean surface-distance MAD (SD) values of 0.17 mm (0.28) and 0.09 mm (0.12), respectively.

Spatial division of the condylar head, the glenoid fossa and the joint space into the four sub-regions, resulted in an increased range of the inter-observer ICCs from good to excellent, (0.96–1.00), (0.71–0.86) and (0.89–0.97), and an increased range of the MAD (SD) values, (2.32% (2.13)–5.01% (3.85)), (0.05 mm (0.05)–0.08 mm (0.06)) and (0.12 mm (0.09)–0.19 mm (0.18)), respectively.

Figure 3 shows Bland–Altman plots of the inter-observer agreement on the measurements. For Bland-Altman plots of the inter-observer agreement on the measurements of the spatial divided condylar head, glenoid fossa and joint space, please refer to Appendix A. The plots show a high degree of consistency in form of low biases, low limits of agreement and few outliers.

## 4. Discussion

The purpose of the present study was to propose and assess the reliability of a semi-automatic approach for holistic 3D assessment of the TMJ from CBCT following orthognathic surgery, including condylar and glenoid fossa changes, as well as the interrelated positional changes. The null hypothesis was: H0: it is not possible to develop a reliable holistic approach for 3D assessment of the TMJ. The approach was deemed reliable if the ICC was good (>0.60), according to Cicchetti [24].

The null hypothesis was rejected by the assessment, showing that the proposed approach has good to excellent inter-observer reliability. When assessing the TMJ without spatial division, the inter-observer reliability was excellent for 3D assessment of all three adaptive processes. When spatially dividing the TMJ into the four sub-regions, the inter-observer reliability remained excellent for 3D assessment of condylar and the interrelated positional changes, whilst the 3D assessment of glenoid fossa changes was excellent for the anterior regions and good for the posterior regions. Both mean- and RMS surface distances were reported in order to demonstrate the reliability of both measures. The minimum or shortest distance has generally been applied in radiographic studies of the TMJ space for decades [26].

The Bland-Altman plots (Figure 3) showed a high degree of observer agreement with only a single outlier in the measurement of condylar and joint space changes. This outlier was caused by a modest inter-observer discrepancy in the assessment of a subject with a high degree of condylar resorption. This tendency seems to be more evident for the mean surface distance measurements of condylar changes, while assessment of subject little to no condylar resorption showed a very high inter-observer agreement. Similarly, the Bland-Altman plots of the spatial 3D assessment (Appendix A) showed a high degree of inter-observer agreement, however, a few outliers were present in the inter-observer measurements. These outliers were mainly caused by an inter-observer discrepancy in the segmentation and the registration results.

The roof of the glenoid fossa is very thin and can cause discontinuities in 3D CBCT scans due to low contrast, spatial resolution, noise, and artefacts, making it difficult to accurately segment and evaluate the fossa [27]. The present approach applied a semi-automatic method for segmentation and 3D reconstruction of the glenoid fossa [4]. Often the glenoid fossa cannot be completely segmented using global image segmentation techniques, and manual segmentation is generally time-consuming and subject to operator variation. For the assessment of relatively small morphovolumetrical changes, such variation affects the reliability of the assessment.

The present study proposed a reliable holistic approach, which can be applied in future clinical studies for semi-automatic 3D evaluation of the TMJ, involving spatial assessment of morphovolumetrical changes, e.g., following orthodontic treatment, orthognathic surgery, trauma, general disease such as rheumatoid arthritis, or to study mandibular growth or asymmetry. The assessment focused on the reliability of the approach. The reliability of the present 3D assessment of condylar changes is in line with the related studies by Verhelst et al. [23] and Xi et al. [28], who reported similar ICC and MAD values.

The accuracy of the 3D assessment of condylar changes from CBCT has been studied previously and the assessment was shown to be highly accurate for both volumetric and surface discrepancy measurements [23]. Similarly, superimposition on the cranial base and the mandibular ramus has shown high accuracy [19,20]. In the study by Holte et al. [20], it was shown that SBR on the mandibular ramus was accurate and reliable for long-term 3D assessment of condylar remodeling following orthognathic surgery, and the measurement error was considered clinically irrelevant. On the contrary, manual landmark-based 3D cephalometric analysis was shown to accumulate landmark identification errors, ranging from 0.02 to 2.47 mm [29,30,31]. Such errors influence the quality of the measurements and make the assessment costly in terms of manual processing time.

Holte et al. also compared the performance of SBR and VBR on the mandibular ramus for long-term 3D assessment of condylar remodeling following orthognathic surgery [20]. The comparative study showed that SBR was more accurate and reliable than VBR using the reference structures proposed by Verhelst et al. [23]. However, it was concluded that the performance of the registration on the mandibular ramus may have been compromised by inappropriate reference structures proposed in the literature, which seem to alter due to remodeling, and may have influenced the registration [20]. Related studies comparing the performance of VBR and SBR for assessment of the outcome of orthognathic surgery [12] and in adult orthodontic subjects [18] found the two registration methods to be accurate and reliable. Voxel-based registration was associated with less variability [12] and was more efficient than SBR [18]. However, the statistical differences between the two registration methods were insignificant and were unlikely to have any clinical significance [12,18].

Several methods have been proposed for the 3D assessment of condylar changes following orthognathic surgery [1,2,3,23,28,32]. However, most of the previous studies on resorption and remodeling of the mandibular fossa relied on a two-dimensional (2D) assessment of lateral cephalograms [15]. Before the introduction of CBCT, 2D transcranial radiography was the standardized imaging modality for studying TMJ [26,33]. However, it is inadequate to describe the complex 3D process of the TMJ remodelling caused by treatment [5,6]. The risk of multiple errors with 2D imaging exists, such as distortion-, magnification-, positioning errors (patient) and overlapping anatomic structures, making landmark identification difficult and cephalometry insufficient to explain the complex 3D process. However, it should be noted that the 3D analysis of the TMJ from CBCT is limited to the assessment of osseous changes, while magnetic resonance imaging enables the assessment of the soft tissue components of the TMJ, i.e., the articular disc, the synovial membrane and the lateral pterygoid muscle [33,34,35].

Recently, fully automated image segmentation of the craniomaxillofacial bones from CBCT has been proposed [36,37]. These studies applied artificial intelligence, more specifically deep learning by neural networks, thereby eliminating the need of manual image segmentation and operator variability [36,37]. Artificial intelligence has been applied for automatic segmentation of the mandibular ramus and condyle [38]. The authors concluded that their findings suggest that CBCT image segmentation of the mandibular ramus and condyle from different clinical centers can be analyzed effectively [38]. Hence, in future studies the semi-automatic segmentation of the mandibular condyle applied in the present study may be substituted with a fully automatic segmentation using artificial intelligence.

To the best of the authors’ knowledge, this is the first study to propose and assess the reliability of a holistic 3D assessment of the TMJ including all three adaptive processes that are believed to contribute to the position of the mandible [21]. Optionally, the proposed method can be combined with the approach by Holte et al [9] in order to include the 3D translational and rotational condylar displacement with six degrees of freedom. The application of a holistic approach is important in order to achieve a more complete understanding, e.g., of the postoperative stability and functionality following orthognathic surgery. Validation of the approach is important in order to obtain accurate and reliable measurements. A semi-automatic implementation of the 3D assessment enables standardization of the measurements, which is important in order to couple and compare results for more conclusive evidence and recommendations based on future meta-analyses. The importance of automation and standardization was concluded in the recent systematic literature review and meta analysis on 3D accuracy and stability of personalized implants in orthognathic surgery, where great heterogeneity was observed in the assessment methods applied in the reviewed literature [39]. Hence, a meta-analysis could only be performed on a sparse subset of the identified studies [39].

## 5. Conclusions

The proposed semi-automatic approach was shown to have good to excellent reliability. It facilitates a holistic 3D quantification and assessment of the TMJ from CBCT, including all three adaptive processes in the TMJ, which are believed to contribute to the position of the mandible: (1) adaptive condylar changes, (2) glenoid fossa changes, and (3) condylar positional changes within the fossa. Application of a standardized holistic approach is important in order to study the TMJ, e.g., to achieve a more complete understanding of its response to orthodontic treatment, orthognathic surgery, trauma, general disease such as rheumatoid arthritis, or to study mandibular growth or asymmetry.

## Figures and Tables

**Figure 1 jpm-13-00343-f001:**
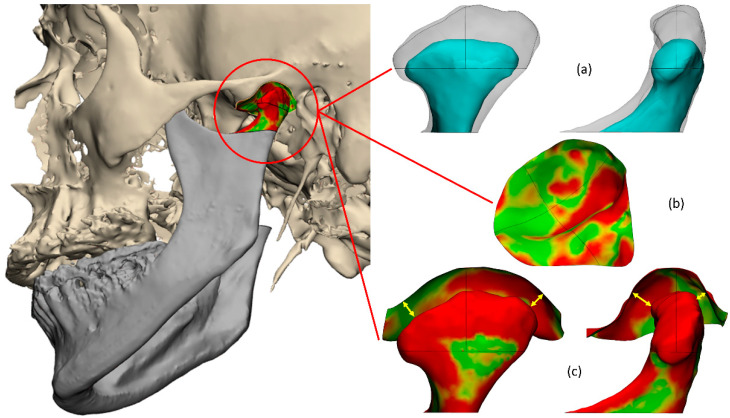
Three-dimensional assessment of morphovolumetrical changes of the TMJ, including all three adaptive processes, which are believed to contribute to the position of the mandible: (**a**) Volumetric condylar changes; (**b**) Three-dimensional fossa changes visualized by color-coded distance maps; (**c**) Minimum 3D joint space distances.

**Figure 2 jpm-13-00343-f002:**
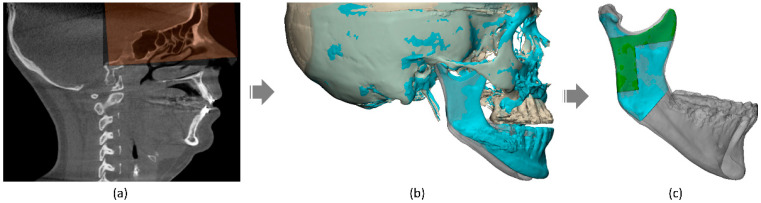
Semi-automation of the 3D assessment using superimposition: (**a**) Voxel-based registration on the anterior cranial base; (**b**) Resulting alignment of the pre- and postoperative cranial 3D models; (**c**) Surface-based registration on the mandibular rami using the stable reference structure proposed by Verhelst et al. [23].

**Figure 3 jpm-13-00343-f003:**
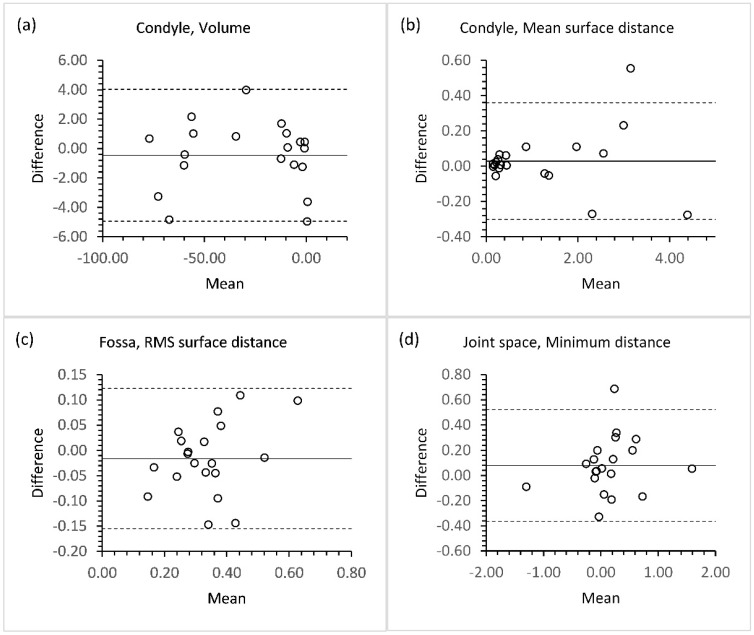
Bland-Altman plots of inter-observer agreement on the 3D TMJ assessments of: (**a**) Volumetric condylar changes; (**b**) condylar changes measured by mean surface distance; (**c**) fossa changes measured by root-mean-squre (RMS) surface distance; (**d**) the minimum 3D joint space distance.

**Table 1 jpm-13-00343-t001:** Intraclass correlation coefficients (ICC), mean absolute differences (MAD) and standard deviations (SD for inter-observer measurements of volumetric condylar remodeling.

	Volumetric Change (%)	
	ICC	MAD (SD)
Condyle	1.00	1.68 (1.58)
Condylar neck	0.98	2.26 (2.04)
Condylar head	1.00	2.71 (2.73)
Anterior-lateral	0.99	3.88 (4.69)
Anterior-medial	0.99	5.01 (3.85)
Posterior-lateral	1.00	2.32 (2.13)
Posterior-medial	0.99	3.29 (2.99)

**Table 2 jpm-13-00343-t002:** Intraclass correlation coefficients (ICC), mean absolute differences (MAD) and standard deviations (SD for inter-observer distance measurements of condylar and glenoid fossa remodeling and changes to the temporomandibular joint space.

	Mean Surface Distance (mm)	
	ICC	MAD (SD)
Condyle	0.99	0.10 (0.14)
Condylar neck	0.96	0.09 (0.12)
Condylar head	0.99	0.17 (0.28)
Anterior-lateral	0.99	0.19 (0.28)
Anterior-medial	0.99	0.15 (0.20)
Posterior-lateral	0.96	0.25 (0.46)
Posterior-medial	0.99	0.14 (0.19)
	**RMS surface distance (mm)**	
	**ICC**	**MAD (SD)**
Glenoid fossa	0.82	0.06 (0.04)
Anterior-lateral	0.86	0.05 (0.05)
Anterior-medial	0.86	0.06 (0.05)
Posterior-lateral	0.71	0.08 (0.06)
Posterior-medial	0.77	0.07 (0.08)
	**Change in minimum joint space distance (mm)**	
	**ICC**	**MAD (SD)**
TMJ joint space	0.91	0.17 (0.16)
Anterior-lateral	0.97	0.12 (0.09)
Anterior-medial	0.89	0.19 (0.16)
Posterior-lateral	0.97	0.15 (0.10)
Posterior-medial	0.92	0.19 (0.18)

RMS: Root mean square.

## Data Availability

Not applicable.

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
