# Peer review of "A Semi-Automatic Approach for Holistic 3D Assessment of Temporomandibular Joint Changes"

_jpm, 2023, doi:10.3390/jpm13020343_

Round 1

Reviewer 1 Report

Comments and Suggestions for Authors

The paper aimed to validate an approach to assess the TMJ following surgery. 

I find the paper rather confusing. The method proposed to have a method that can compare the TMJ changes following surgery. Firstly, this semi-automatic method was not described in the paper but the author expected the reader to search and read up on such method. It would be good if it was supplemented with images of how the methods were done. How much the method mimics that of the cited paper was not clear. The author attempted to assess the validity by comparing the measurements made by two different investigators. However, the author should have described what is the current acceptable or gold method. If not this should be compared with a model first. To determine if a new method is acceptable, one needs to be clear on the concept of validity and reliability. Then one can compare the accepted/gold method with the new method. One should also then compare the inter and intra - rater reliability of such new method. 

Overall, I was not convinced if this method was valid and reliable.

Author Response

We thank the reviewer for the suggestions to improve the manuscript and for taking the time to review our submission.

1. Firstly, this semi-automatic method was not described in the paper but the author expected the reader to search and read up on such method. It would be good if it was supplemented with images of how the methods were done.

We have elaborated on the semi-automatic method in the method section and included a new Figure 2 illustrating the process.

2. How much the method mimics that of the cited paper was not clear.

This is the first time a holistic approach is proposed that constitutes all three adaptive processes in the TMJ that are believed to contribute to the position of the mandible. The cited paper [4] solely presented a clinical evaluation of mandibular glenoid fossa changes following orthognathic surgery. Hence, a clinical study and not a method study, including only one of the adaptive processes.

The other cited paper [19] presented a comparative study on two different registration techniques. The findings of this paper were used in the present study for selection of the best technique for registration on the mandibular ramus in the semi-automatic method. This has been elaborated in the method section:

“Surface-based registration was used for the alignment, which was shown to be more accurate and reliable than voxel-based registration for assessment of mandibular condyle remodeling [19].”

3. The author attempted to assess the validity by comparing the measurements made by two different investigators. However, the author should have described what is the current acceptable or gold method. If not this should be compared with a model first. To determine if a new method is acceptable, one needs to be clear on the concept of validity and reliability. Then one can compare the accepted/gold method with the new method. One should also then compare the inter and intra - rater reliability of such new method.

We acknowledge the reviewers comment on validity. We have discussed how the accuracy of the image segmentation and 3D reconstruction of the mandibular condyles has previously been validated in recent literature [22]. To this end, the accuracy of the superimposition techniques which were used in the semi-automatic implementation of the method has been previously validated in [18,19]. Hence, the present study focused on the reliability of the new holistic approach. This has been clarified in the introduction:

“No studies have been identified, which propose a validated holistic approach for 3D assessment of the TMJ including all three adaptive processes. However, the accuracy of the 3D assessment of condylar changes from CBCT has been studied previously and the assessment was shown to be highly accurate for both volumetric and surface discrepancy measurements [22]. Similarly, superimposition on the cranial base and the mandibular ramus has shown high accuracy [18,19]. Hence, the purpose of the present study was to propose and validate the reliability of a semi-automatic approach for 3D assessment of the TMJ from cone-beam computed tomography (CBCT) following Orthognathic Surgery, including condylar and glenoid fossa changes, as well as the interrelated positional changes.”

Again, we thank the reviewer for the constructive feedback to improve our manuscript, and sincerely hope the revision meets the expectations of the reviewer.

Reviewer 2 Report

Comments and Suggestions for Authors

Dear authors, your paper underwent an additional review.

Congrats, this is a fine and scientifically sound paper, which I enjoyed reading. What may be added (viz. is unluckily missing) is a "limitations of the study" section. It should be addressed (which in fact is cursorily mentioned in the methods section) that the CBCTS were not - or at least may not have been - taken in a reproducible (viz. standardized) condylar position; CBCT pre-op was taken in a wax bite position (it remains unclear whether under maximum pre-op dental contact or in centric or any therapeutic relation?), post-op CBCT, however, without any such a specified position. Therefore, the patients may have opened their mouth, may have gone into a painfree position or have been in maximum intercuspidation etc.  This will or at least may lead to potential 3D positional (viz. erroneously) assessed measurements (which may show excellent interrater results, but are clinically not comparable and valid with regard to the goal of a holistic 3D bony assessment of the TMJ, which is the primary goal of the paper (i.e., it is not accuracy of measurements, alone). Rather surprisingly, the condylar position does not alter much, as is usually the case after mandibular advancement.

To sum up, the underlying problem of a 3D bony assessment of condylar position requires additional measures to warrant reproducibility and this is not described in the paper, yet. 

I hope the authors take up the burden (or chance) to add such a limitations section and address this criticism in the discussion

Round 2

Reviewer 1 Report

Comments and Suggestions for Authors

Thank you for the improvement of the paper. However, the authors should still be careful when using the terms validity and reliability. Since the study did not compare the new method with another method (ideally a gold standard), then one should avoid claiming for validity. The study only compared the reliability/reproducibility of its method. Hence, my suggestion would be to change the abstract :

1) The literature lacks a validated holistic approach for three-dimensional (3D) assessment of the temporomandibular joint (TMJ)... to The literature lacks a reliable holistic approach for three-dimensional (3D) assessment of the temporomandibular joint (TMJ)..

2) Hence, the purpose of the present study was to propose and validate a semi-automatic approach for 3D assessment of the TMJ from cone-beam computed tomography (CBCT) following oOrthognathic sSurgery... to Hence, the purpose of the present study was to propose and assess the reliability of a semi-automatic approach for 3D assessment of the TMJ from cone-beam computed tomography (CBCT) following oOrthognathic sSurgery. Similarly in the introduction, the aims should be modified accordingly.

Author Response

We thank the reviewer for the additional suggestions to improve the manuscript and for taking the time for a second round of reviewing our submission.

Thank you for the improvement of the paper. However, the authors should still be careful when using the terms validity and reliability. Since the study did not compare the new method with another method (ideally a gold standard), then one should avoid claiming for validity. The study only compared the reliability/reproducibility of its method. Hence, my suggestion would be to change the abstract :

1) The literature lacks a validated holistic approach for three-dimensional (3D) assessment of the temporomandibular joint (TMJ)... to The literature lacks a reliable holistic approach for three-dimensional (3D) assessment of the temporomandibular joint (TMJ)..

2) Hence, the purpose of the present study was to propose and validate a semi-automatic approach for 3D assessment of the TMJ from cone-beam computed tomography (CBCT) following oOrthognathic sSurgery... to Hence, the purpose of the present study was to propose and assess the reliability of a semi-automatic approach for 3D assessment of the TMJ from cone-beam computed tomography (CBCT) following oOrthognathic sSurgery. Similarly in the introduction, the aims should be modified accordingly.

We agree that solely the reliability/reproducibility of the proposed method was assessed in the study. The abstract, the aims in the introduction, and the discussion have been changed according to the reviewer’s suggestions.

We sincerely hope the minor revision meets the expectations of the reviewer.